# Scratch-AID, a deep learning-based system for automatic detection of mouse scratching behavior with high accuracy

Huasheng Yu[1]*[†], Jingwei Xiong[2][†], Adam Yongxin Ye[3], Suna Li Cranfill[1], Tariq Cannonier[1], Mayank Gautam[1], Marina Zhang[4], Rayan Bilal[1], Jong-Eun Park[1], Yuji Xue[1], Vidhur Polam[1], Zora Vujovic[1], Daniel Dai[1], William Ong[1], Jasper Ip[1], Amanda Hsieh[1], Nour Mimouni[1], Alejandra Lozada[1], Medhini Sosale[1], Alex Ahn[1], Minghong Ma[1], Long Ding[1], Javier Arsuaga[5,6], Wenqin Luo[1]*

[1]Department of Neuroscience, Perelman School of Medicine, University of Pennsylvania, Philadelphia, United States; [2]Graduate Group in Biostatistics, University of California Davis, Davis, United States; [3]Howard Hughes Medical Institute, Program in Cellular and Molecular Medicine, Boston Children's Hospital, Harvard Medical School, Boston, United States; [4]Department of Electrical Engineering and Computer Science, Massachusetts Institute of Technology, Cambridge, United States; [5]Department of Molecular and Cellular Biology, University of California Davis, Davis, United States; [6]Department of Mathematics, University of California Davis, Davis, United States

*For correspondence:
huasheng.yu@pennmedicine.
upenn.edu (HY);
luow@pennmedicine.upenn.edu
(WL)

[†]These authors contributed
equally to this work

Competing interest: The authors
declare that no competing
interests exist.

Reviewing Editor: Brian S Kim,
Icahn School of Medicine at
Mount Sinai, United States

**Abstract** Mice are the most commonly used model animals for itch research and for development of anti-itch drugs. Most laboratories manually quantify mouse scratching behavior to assess itch intensity. This process is labor-intensive and limits large-scale genetic or drug screenings. In this study, we developed a new system, Scratch-AID (Automatic Itch Detection), which could automatically identify and quantify mouse scratching behavior with high accuracy. Our system included a custom-designed videotaping box to ensure high-quality and replicable mouse behavior recording and a convolutional recurrent neural network trained with frame-labeled mouse scratching behavior videos, induced by nape injection of chloroquine. The best trained network achieved 97.6% recall and 96.9% precision on previously unseen test videos. Remarkably, Scratch-AID could reliably identify scratching behavior in other major mouse itch models, including the acute cheek model, the histaminergic model, and a chronic itch model. Moreover, our system detected significant differences in scratching behavior between control and mice treated with an anti-itch drug. Taken together, we have established a novel deep learning-based system that could replace manual quantification for mouse scratching behavior in different itch models and for drug screening.

## Editor's evaluation

Scratch assays are the gold standard for measuring itch in rodents. However, the current limitation is that this is performed manually which is enormously taxing in terms of hours spent counting scratching bouts. The authors have developed a valuable automatic system to quantify scratch behavior with high accuracy and provided a valuable tool for the field. This will be resourceful for the greater itch biology community.

## Introduction

Itch is a disturbing symptom associated with skin diseases, immune problems, systemic diseases, and mental disorders (*Cevikbas and Lerner, 2020*; *Hong et al., 2011*; *Kremer et al., 2020*; *Ständer et al., 2007*). Chronic itch affects about 13–17% of the population (*Matterne et al., 2009*; *Weisshaar and Dalgard, 2009*), severely worsening the quality of life of affected patients. Unfortunately, treatment options for many chronic itch conditions are still limited (*Yosipovitch et al., 2018*; *Yu et al., 2021*).

Mice are the most widely used model animals for studying itch mechanisms and for developing new preclinical anti-itch drugs (*Han et al., 2013*; *Liu et al., 2009*; *Solinski et al., 2019*; *Sun and Chen, 2007*). Since itch is an unpleasant sensation that provokes the desire to scratch (*Ikoma et al., 2006*), scratching behavior has been assessed as a proxy for itch intensity in mice (*Liu et al., 2009*; *Morita et al., 2015*). Till now, this quantification process has been mainly conducted by watching videos and manually counting scratching bouts or the total scratching time, which is tedious and time consuming, unavoidably introduces human errors and bias, and limits the large-scale genetic or drug screenings.

Given the biological importance and the obvious need, several research groups have tried different strategies to automate this process, including an acoustic recording method (*Elliott et al., 2017*), a method using magnetic field and metal ring to detect paw movement (*Mu et al., 2017*), and several video analysis-based approaches (*Bohnslav et al., 2021*; *Kobayashi et al., 2021*; *Park et al., 2019*; *Sakamoto et al., 2022*). Nevertheless, these methods have not been widely adopted by other research laboratories, due to the uncertain performance of the trained models in different lab environments, the requirement of specialized equipment, and/or inadequate evaluation of these methods in different mouse itch models.

In recent years, with the rapid progress in the field of artificial intelligence, deep learning has been applied in various scientific research areas. For example, convolutional neural networks (CNN) are widely used in computer visual recognition tasks (*Gu et al., 2018*), whereas recurrent neural networks (RNN) (*Graves, 2013*) are developed for analyzing temporal dynamic features. Moreover, rapid improvement of computing power, especially in the graphics processing unit capacity, together with new open-source deep learning libraries, such as PyTorch (*Paszke et al., 2019*), Keras (*Gulli and Pal, 2017*), and Tensorflow (*Abadi et al., 2016*), have greatly accelerated broad applications of deep learning.

Animal behavior analysis is one of the research areas benefiting from the applications of deep learning. For example, DeepLabCut can track different body parts in freely moving animals for behavior analysis (*Mathis et al., 2018*). DeepEthogram recognizes and annotates different behaviors of mice and flies (*Bohnslav et al., 2021*). These examples support the proof-of-principle that deep-learning is a powerful avenue for automating animal behavior analysis. Nevertheless, for a given animal behavior, like mouse scratching, a designated method, which achieves high sensitivity, specificity, and generalization to replace human observers, still needs to be established.

To meet this challenge, we developed a new deep learning-based system, Scratch-AID (**A**utomatic **I**tch **D**etection), which achieved automatic quantification of mouse scratching behavior with high accuracy. We first designed a videotaping box to acquire high-quality recording of mouse behavior in a reproducible environment from the bottom. We recorded 40 videos of 10 adult wildtype mice (5 males and 5 females) after nape injection of a non-histamine pruritogen, chloroquine (CQ), and manually labeled all video frames as the reference. We then designed a convolutional recurrent neural network (CRNN) by combining CNN and RNN and trained it with 32 scratching videos from 8 randomly picked mice. We obtained a series of prediction models using different training parameters and evaluated these models with test videos (eight unseen test videos from the two remaining mice). The best trained model achieved 97.6% recall and 96.9% precision on test videos, similar to the manual quantification results. Impressively, Scratch-AID could also quantify scratching behavior from other major acute and chronic itch models with high accuracy. Lastly, we applied Scratch-AID in an anti-itch drug screening paradigm and found that it reliably detected the drug effect. In summary, we have established a new system for accurate automatic quantification of mouse scratching behavior. Based on the performances, Scratch-AID could replace manual quantification for various mouse itch models and for drug or genetic screenings.

## Results

### The overall workflow

Our workflow to develop a new system for detection and quantification of mouse scratching behavior consists of four major steps (*Figure 1A*): (1) Videotape mouse scratching behavior induced by an acute nape itch model; (2) Manually annotate scratching frames in all videos for training and test datasets; (3) Design a deep learning neural network; train this network with randomly selected training videos and adjust different training parameters; and evaluate the performance of the trained neural networks on test videos; (4) Evaluate the generalization of the trained neural network in additional itch models and a drug screening paradigm.

### Design a videotaping box for high-quality and reproducible recording of mouse scratching behavior

High-quality videos recorded from a reproducible environment are critical for stable performance of trained prediction models and for adoption by other research laboratories. Thus, we designed a mouse videotaping box for such a purpose. It consisted of two boxes with white acrylic walls joined by a transparent acrylic floor (*Figure 1B*). The top ('mouse') box (Length × Width × Height = 14.68 × 14.68 × 5 cm) had a lid for mouse access. The bottom ('camera') box (Length × Width × Height = 14.68 × 14.68 × 23.6 cm) had a door for access to the camera (Logitech C920e Business Webcam). The walls and the lid of the box were non-transparent to minimize interference from outside visual stimuli. Ambient light penetrated the walls and provided sufficient illumination for behavior recording. A mouse could freely move inside the top box, and a camera recorded mouse behavior from the bottom (30 frames/s). Compared to the top or side views, the bottom view can clearly capture the key body parts involved in scratching behavior, such as the scratching hind paw and mouth, and their movements in great details (*Video 1*). The magnification, resolution, and brightness of the video can be adjusted by the camera recording software (Logitech C920e Business Webcam driver and software) to achieve consistent video recording. In short, this customized videotaping box allows high-quality video recording of mouse scratching and other behaviors in a stable and reproducible environment.

Spontaneous scratching is a rare event under normal conditions in mice. Itch sensation and scratching are usually induced by different itch models for research. Common mouse itch models are classified as cheek or nape, histaminergic or non-histaminergic, and acute or chronic, based on the body location where itch sensation is evoked, the kind of pruritogens, and the duration of itch sensation (*Ikoma et al., 2006*; *Liu and Dong, 2015*; *Shimada and LaMotte, 2008*; *Thurmond et al., 2008*). We first used an acute nape itch model induced by a non-histaminergic pruritogen, CQ, because it triggered immediate and robust scratching behavior in mice (*Liu et al., 2009*). After intradermal injection of CQ (200 µg in 15 µl saline) into left or right nape of the mice, a 20-min video was recorded using the customized videotaping box (*Figure 1C*).

In response to CQ injection, mice scratched the affected skin region using their ipsilateral hind paw. Evoked mice displayed multiple episodes of scratching, which were separated by non-scratching phases. Each episode of scratching, here defined as a scratching train, usually contained four phases: start (lifting the scratching hind paw toward the affected skin), scratching bout (rhythmic movement of hind paw against the affected skin), paw licking (putting the scratching hind paw into the mouth and licking), and end (putting down the scratching hind paw back to the floor) (*Figure 1D* and *Video 1*). The cycle of scratching bout and paw licking might occur once or repeat multiple times, depending on the itch intensity and the internal state of the mouse. The time from the start to the end of a given scratching train is defined as the duration. The total scratching time is the sum of durations of all scratching trains, which is an effective parameter to quantify the scratching behavior and assess itch intensity.

### Video annotation

Forty scratching videos from 10 adult wildtype C57 mice (5 males and 5 females, 2- to 3-month-old) were recorded (*Supplementary file 1*). For neural network training and testing and for comparing performances between trained neural networks and manual quantification, two methods were used to annotate mouse scratching behavior in these videos. The first method (*manual annotation*) was to watch these videos at normal (1×) speed (30 frames/s) and label the start and end time points

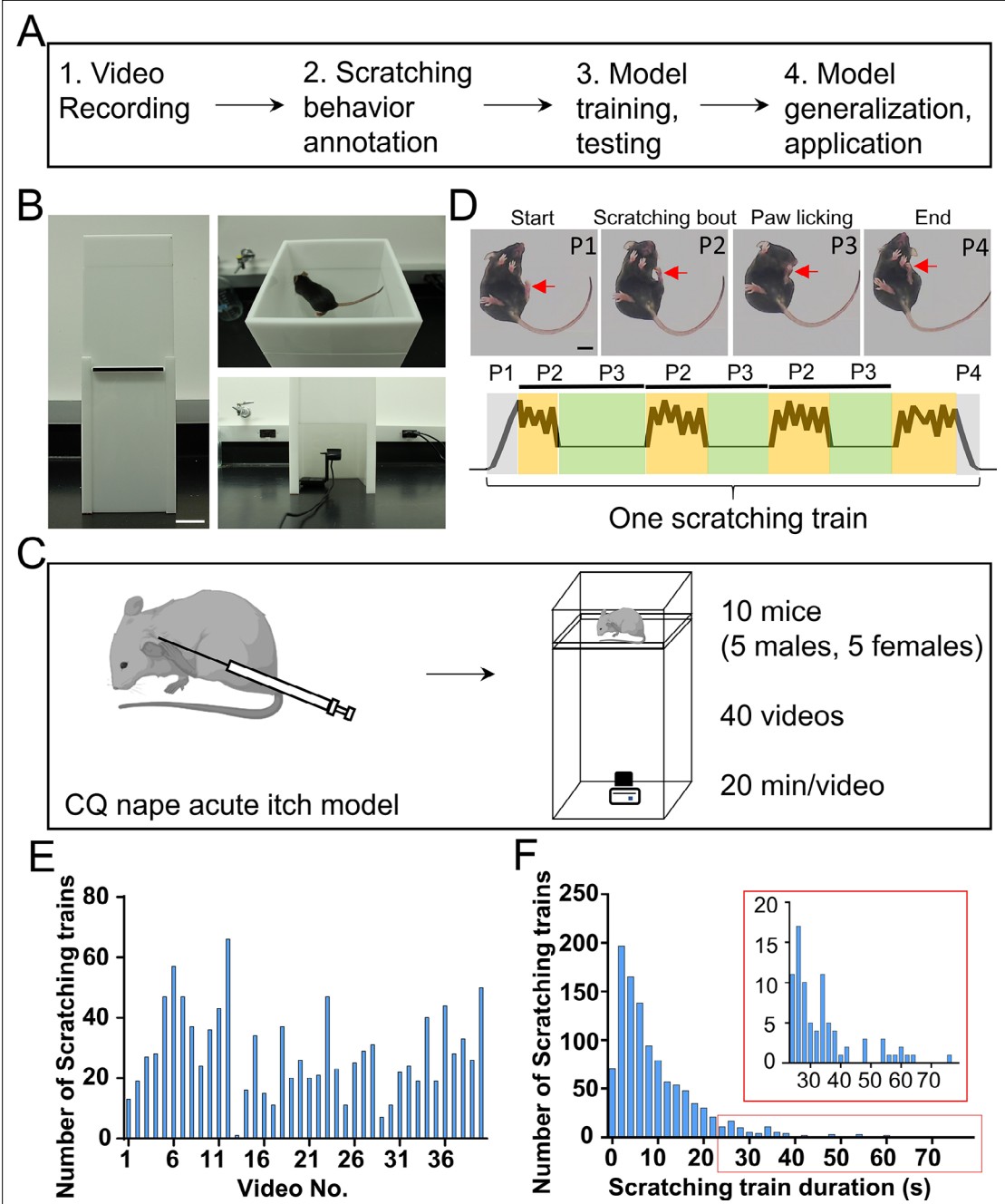

**Figure 1.** The overall workflow and building a customized videotaping box for mouse scratching behavior recording. (**A**) A diagram showing the workflow to develop a deep learning-based system for automatic detection and quantification of mouse scratching behavior. (**B**) An image of the designed videotaping box for high-quality video recording of mouse scratching behavior. Scale bar, 5 cm. (**C**) A cartoon showing the acute itch model induced by the chloroquine (CQ) injection in the nape, followed by video recording in the customized videotaping box. (**D**) Representative images showing different phases (P1–P4) of a scratching train (upper). Red arrows indicate the scratching hind paw. A cartoon showing the dynamic movement of the scratching hind paw in a scratching train (bottom). The cycle of scratching bout (P2) and paw licking (P3) may repeat once or more times in a scratching train. Scale bar, 1 cm. (**E**) The total number of scratching trains in each video. (**F**) The distribution of scratching train duration ($n$ = 1135 scratching trains). The inset is the zoom-in of the red rectangle.

The online version of this article includes the following figure supplement(s) for figure 1:

**Figure supplement 1.** Dynamic and static features of scratching behavior in the chloroquine (CQ) nape acute itch model.

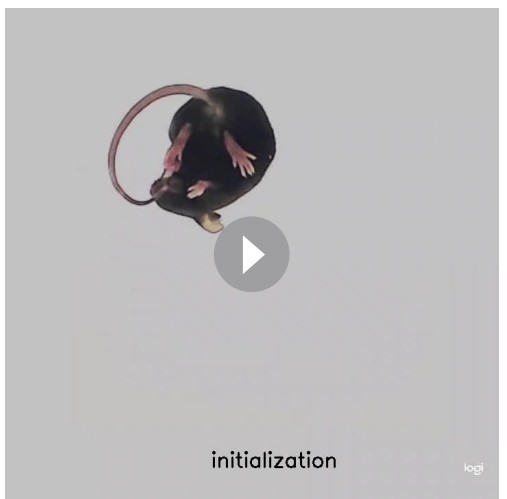

**Video 1.** An example of a mouse scratching train recorded by the designed videotaping box.
https://elifesciences.org/articles/84042/figures#video1

(converted into frame numbers for subsequent analysis) of each scratching train. This is consistent with the field common practice for manually quantifying mouse scratching behavior. Our manual annotation results were produced by 10 human observers, thus reflecting an averaged precision of the manual quantification process. The second method (*reference annotation*, or ground-truth annotation) was to accurately determine the start and end of each scratching train by analyzing each video frame-by-frame. The reference annotation of the 40 videos were used as the training and test datasets. The total number of scratching train in each video and the distribution of scratching train durations were quantified (*Figure 1E, F* and *Supplementary file 2*).

## Deep learning neural network design and model training

Mouse scratching behavior displayed unique dynamic (temporal) and static (spatial) features, which were highlighted by tracking the key body parts using DeepLabCut (*Mathis et al., 2018*; *Figure 1—figure supplement 1*; *Video 2*). One of the most obvious dynamic features was the rhythmic movement of the scratching hind paw (*Figure 1—figure supplement 1A, B*). Some unique static features included the relative positional relationships between the scratching hind paw and other body parts (*Figure 1—figure supplement 1C–F*). To fully capture these dynamic and static features, we designed a CRNN to take advantage of the different strengths of CNN and RNN (*Figure 2A* and *Figure 2—figure supplement 1A*). The CRNN contained a CNN (ResNet-18 *He et al., 2016*; *Figure 2—figure supplement 1B*) that extracts static features, such as the relative position of different body parts, an RNN (two-layer bidirectional gated recurrent unit [GRU]; *Dey and Salem, 2017*; *Figure 2—figure supplement 1C*) that extracts dynamic features, such as the rhythmic movement of the scratching hind paw in consecutive frames, and a fully connected layer (the classifier) to combine the features extracted by both CNN and RNN and generate the prediction output.

The 40 videos were randomly split into two parts, 80% of them (32 videos from 8 mice) were assigned to the training dataset and 20% of them (8 videos from 2 mice) to the test dataset (*Figure 2C*). Each video was converted into individual frames, and each frame was classified as 'scratching' (within a scratching train) or 'non-scratching' (out of a scratching train) based on the reference annotation. For the training dataset preparation, *N* consecutive frames (a parameter adjusted for optimal model performance) were selected as one input to capture the dynamic features of scratching (*Figure 2B*). To avoid large sets of redundancy in the training dataset, two adjacent inputs were apart between 4 and 10 frames (*Figure 2B*). An input was labeled as 'scratching' (class 1) if more than half of frames ($N/2$) in the input were scratching frames; otherwise labeled as 'non-scratching' (class 0) (*Figure 2B*).

Since the dynamic features of scratching behavior spanned multiple frames, the input length (*N* frames) would be a critical training parameter. In CQ triggered acute nape itch

**Video 2.** Key body parts tracked by DeepLabCut showing the dynamic and static features of mouse scratching behavior.
https://elifesciences.org/articles/84042/figures#video2

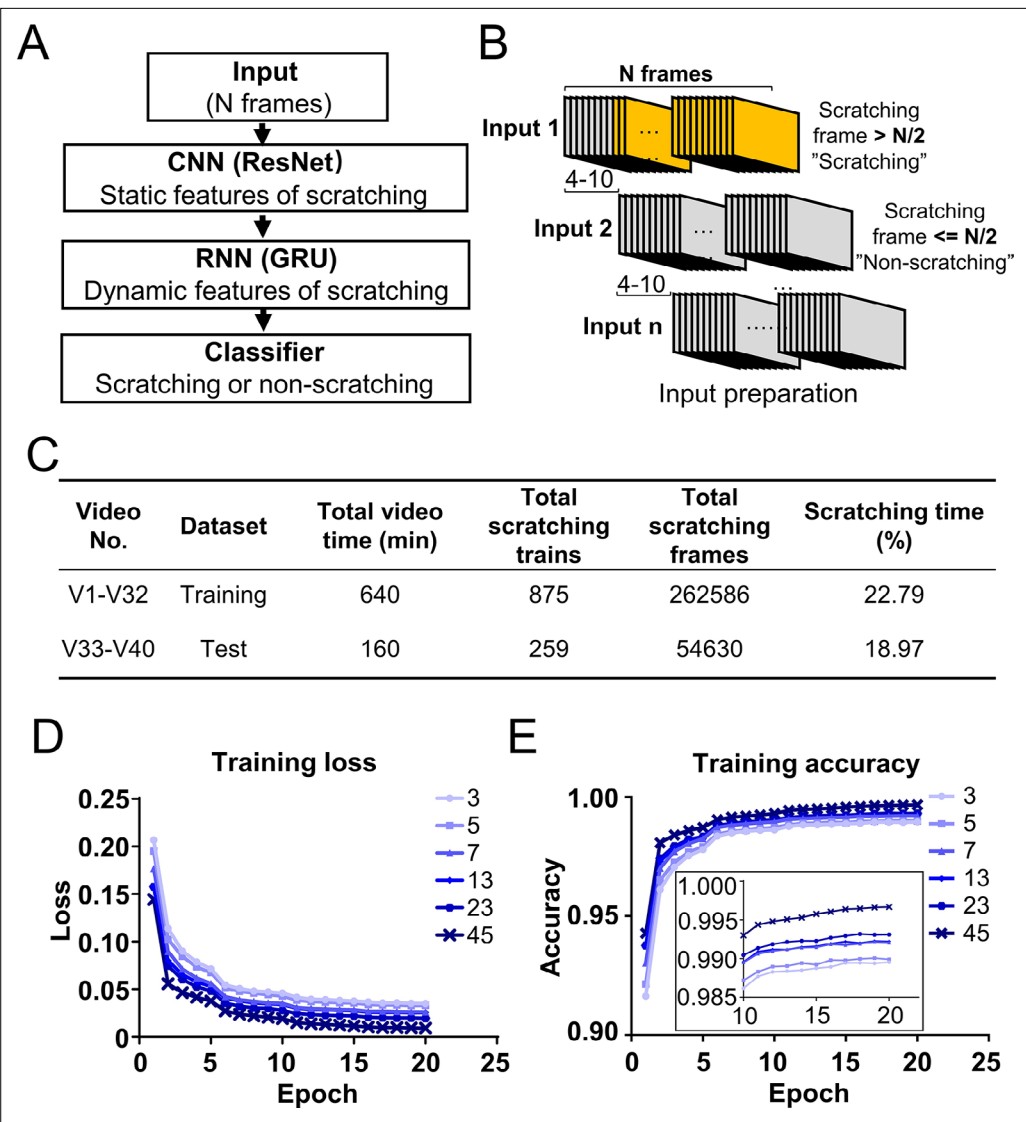

**Figure 2.** Deep learning neural network design and training. (**A**) Cartoon showing the architecture of designed deep learning neural network consisting of the combination of convolutional neural networks (CNN), recurrent neural networks (RNN), and classifier. (**B**) Cartoon showing the preparation of inputs for the training dataset. Consecutive *N* frames were selected as one input for training. The interval between two adjacent inputs in a video was 4–10 frames. (**C**) The information of a sample training and test datasets. The training loss decreased (**D**) while the accuracy increased (**E**) during the training process with different input length (*N* = 3, 5, 7, 13, 23, 45 frames). The inset is the zoom-in of part of the figure.

The online version of this article includes the following figure supplement(s) for figure 2:

**Figure supplement 1.** The architecture of deep learning neural network.

**Figure supplement 2.** Distribution of durations of scratching bout and paw licking in the chloroquine (CQ) nape acute itch model.

model, the average duration of one cycle of scratching bout and paw licking was around 30 frames (*Figure 2—figure supplement 2A, B*). Thus, we tested a range of input length from 3 to 45 frames for model training. During the training process, the loss (discrepancy between model prediction and reference annotation) decreased, and the prediction accuracy (correct prediction of both scratching and non-scratching frames/all frames) increased (*Figure 2D, E*). After 10 epochs (one epoch means the training covers the complete training dataset for one round), the accuracy reached a plateau (*Figure 2E*). The prediction accuracies were more than 0.98 for all input length, improving slightly with the increase of the input length (*Figure 2E*). These results demonstrate that the designed CRNN

network works very efficiently to capture the scratching features and recognize scratching behavior in the training dataset.

## Model evaluation on test datasets

We evaluated the performance of the trained prediction models on eight unseen test videos. First, similar to what described above, each test video was converted into inputs with '*N*' frames (the same '*N*' was used for training and test), except that the two adjacent inputs were only 1 frame apart. Second, the trained neural network predicted each input to be 'scratching' or 'non-scratching'. Third, to convert the prediction from one input (containing *N* frames) into the prediction for each individual frame, we used the following rule: the prediction of the middle frame of each input would be the same as that of the input. For example, if an input was predicted as 'scratching', then the middle frame of this input would be a 'scratching' frame. This conversion predicted each frame of tested videos as 'scratching' or 'non-scratching' expect for the few frames at the beginning or at the end of a video (see method for the missing data interpretation). Fourth, recall (the number of correctly predicted scratching frames/the number of reference scratching frames), precision (the number of correctly predicted scratching frames/the number of all predicted scratching frames), and *F*1 score (2*recall*precision/(recall + precision)) were calculated. Compared to the overall accuracy (correct prediction of both scratching and non-scratching frames/all frames), the recall, precision, and *F*1 score give a more precise and in depth evaluation of a model's performance (*Powers, 2020*), especially when scratching is a relative rare event in a video.

To rule out the possibility that the good performance of our models was due to a specific combination of the training and test datasets, we rotated the training and test videos for cross-validation. For all different combination of training and test videos, *F*1 scores were all above 0.9 (*Figure 3—figure supplement 1A, B*), supporting the stable and high performance of our prediction models. In addition, the prediction model performed better with the increased input length (*Figure 3—figure supplement 1C*). The best model, the one trained with videos 1–32 with the input length of 45 (*N* = 45), was selected for additional analyses and tests.

The average recall and precision of the best model were 97.6% and 96.9%, respectively, for the eight test videos (*Figure 3A*), and the recall and precision for individual videos were above 95% in most cases (*Figure 3B*). This performance was similar to or even slightly better than that of manual annotation, which had an average recall and precision of 95.1% and 94.2%, respectively (*Figure 3C, D*). When comparing the total scratching time to the reference annotation, the average discrepancy of the model prediction was 1.9% whereas that of manual annotation was 2.1% (*Figure 3E*). The correlation between the model prediction and the reference annotation was 0.98, similar to the manual annotation results (*Figure 3F*). When examining the probability traces of model prediction and the reference annotation (one example shown in *Figure 3G*), we found that the model successfully recognized almost all the scratching trains in test videos, and that the prediction of the start and end of each scratching train aligned well with the reference annotation (*Figure 3H*). Taken together, these results demonstrate the high reliability and accuracy of our model to recognize and quantify mouse scratching behavior of new videos.

## The trained neural network model focused on the scratching hind paw to recognize mouse scratching behavior

How did the trained neural network model recognize mouse scratching behavior and distinguish them from other behaviors? Although deep learning neural networks are processed as a black box, saliency maps can give some hints (*Selvaraju et al., 2017*), because they plot which parts of each frame (pixels) were mainly used during model prediction. The most salient parts were centered around the scratching hind paw in the scratching frames (*Figure 4A, B* and *Figure 4—figure supplement 1A*), suggesting that the prediction model focused on the features of the scratching hind paw. In some scratching frames, other body parts were also highlighted, such as the two front paws (*Figure 4B*), suggesting the model also utilized the positional relationship of these body parts to recognize scratching behavior. In contrast, for other 'non-scratching' mouse behaviors, such as wiping, grooming, rearing, and locomotion, the 'salient' parts showed no clear association with particular mouse body parts (*Figure 4C–F* and *Figure 4—figure supplement 1B–E*). Together, these saliency maps indicate that the trained

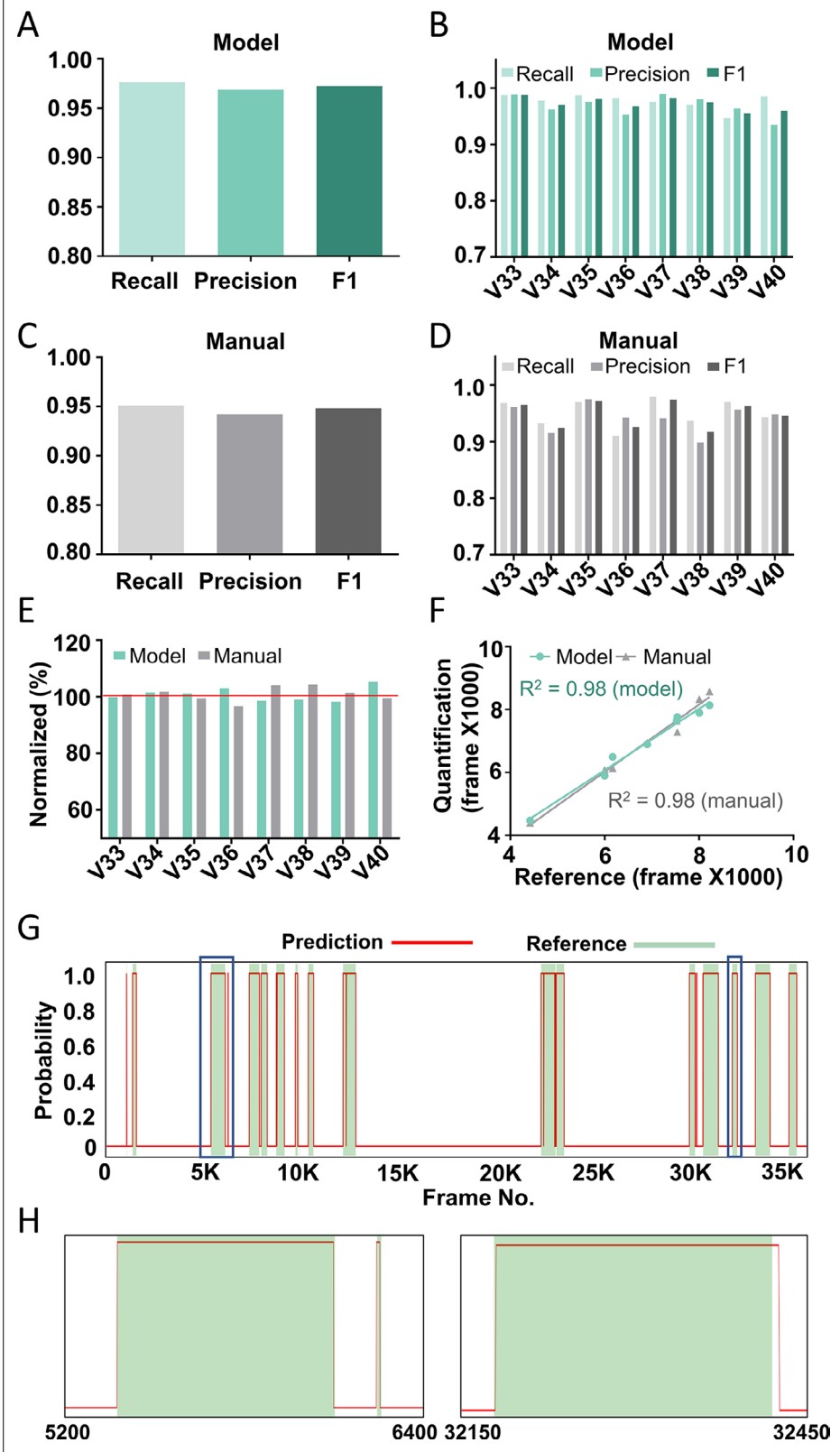

**Figure 3.** Performance of the best model on test videos. The recall, precision, and *F*1 score of the best model on average (**A**) or in individual videos (**B**). The recall, precision, and *F*1 score of manual annotation on average (**C**) or in individual videos (**D**). (**E**) The comparison among model prediction, manual quantification, and the reference annotation. The reference annotation is normalized to 100% shown as the red line. (**F**) The correlations between

*Figure 3 continued on next page*

*Figure 3 continued*

model prediction or manual quantification and reference annotation. $R^2$, Pearson correlation coefficient. (**G**) An example scratching probability trace (red curve) predicted by the model and aligned with the reference annotation (green bar). (**H**) The two zoom-ins from (**G**) showing the nice alignment between the model prediction and the reference annotation.

The online version of this article includes the following figure supplement(s) for figure 3:

**Figure supplement 1.** Cross-validation and parameter optimization of the prediction models.

**Figure supplement 2.** Error analysis of the best prediction model.

**Figure supplement 3.** Error analysis of the manual quantification.

**Figure supplement 4.** Other mouse behaviors were not recognized as scratching behavior.

**Figure supplement 5.** Relationship between prediction errors and the input length or the scratching train duration.

neural network learns to focus on the dynamic and static features of scratching for the prediction of mouse scratching behavior.

## Model prediction error analysis

To further understand the performance of the best trained neural network model, we systematically analyzed its prediction errors in eight test videos (*Figure 3—figure supplement 2*) and compared to those from the manual quantification (*Figure 3—figure supplement 3*) and other trained models. We classified the prediction errors into five categories: Type 1, false positive (non-scratching region was predicted as a scratching train); Type 2, false negative (a real scratching train was not recognized); Type 3, blurred boundary (the prediction of the start or end of a scratching train was shifted); Type 4, missed interval (two or more adjacent scratching trains were predicted as one scratching train); and Type 5 split scratching train (one scratching train was predicted as two or more scratching trains) (*Figure 3—figure supplement 2A*). We found that the dominant prediction error of the trained neural network model was Type 3 error, accounting for around 3% of the total scratching frames, followed by Type 2 and 5 errors accounting for around 1% (*Figure 3—figure supplement 2B*). For manual quantification, the major errors came from Types 3 and 4, accounting for 10% and 8% of the total scratching frames, respectively (*Figure 3—figure supplement 3A*).

For Type 1 error of the model prediction (*Figure 3—figure supplement 2C1–C3*), the durations of all false positive scratching trains were shorter than 10 frames (0.3 s) (*Figure 3—figure supplement 2C2*) and temporarily close to a real scratching train (within 30 frames, <1 s) (*Figure 3—figure supplement 2C3*). They were not caused by confusion with other behaviors, such as wiping, grooming, rearing, locomotion, and resting (*Figure 3—figure supplement 4*). Type 1 error was also rare for manual annotation (*Figure 3—figure supplement 3B*).

The models might miss short scratching trains, hence caused the Type 2 error. Indeed, all missed scratching trains were shorter than 40 frames (<1.3 s) (*Figure 3—figure supplement 2D1, D2*). For all scratching trains lasting less than 30 frames (<1 s), 18.5% of them were missed by the model prediction. This number decreased to 2.7% for scratching trains spanning between 30 and 60 frames (1–2 s). No scratching train was missed if they were longer than 60 frames (>2 s) (*Figure 3—figure supplement 2D3*). The Type 2 error positively correlated with the input length (*N*) of prediction models. It became zero or close to zero when models trained with shorter input lengths (3, 13, and 23 frames) (*Figure 3—figure supplement 5A–C*). For manual annotation, Type 2 error was not common (*Figure 3—figure supplement 3C1, C2*).

Type 3 error (*Figure 3—figure supplement 2E1–E3*) was dominant among all five type errors. The average start and end frame shift of the model prediction were 2.2 and 7.0 frames (*Figure 3—figure supplement 2E3*), while those of the manual annotation were 11.5 and 12.8 frames (*Figure 3—figure supplement 3D1, D2*). The start and end frame shift of the manual quantification was similar (~350 ms), which likely reflected the temporal delay of the real time human visual system processing. For the model prediction, which was an off-line frame by frame process, the temporal shifts were less than the human visual processing. In addition, the start of a scratching train was more accurately recognized by the model than the end of a scratching train (*Figure 3—figure supplement 2E3*). This might reflect the feature of scratching trains. It was relatively clear to determine the start of a scratching train when

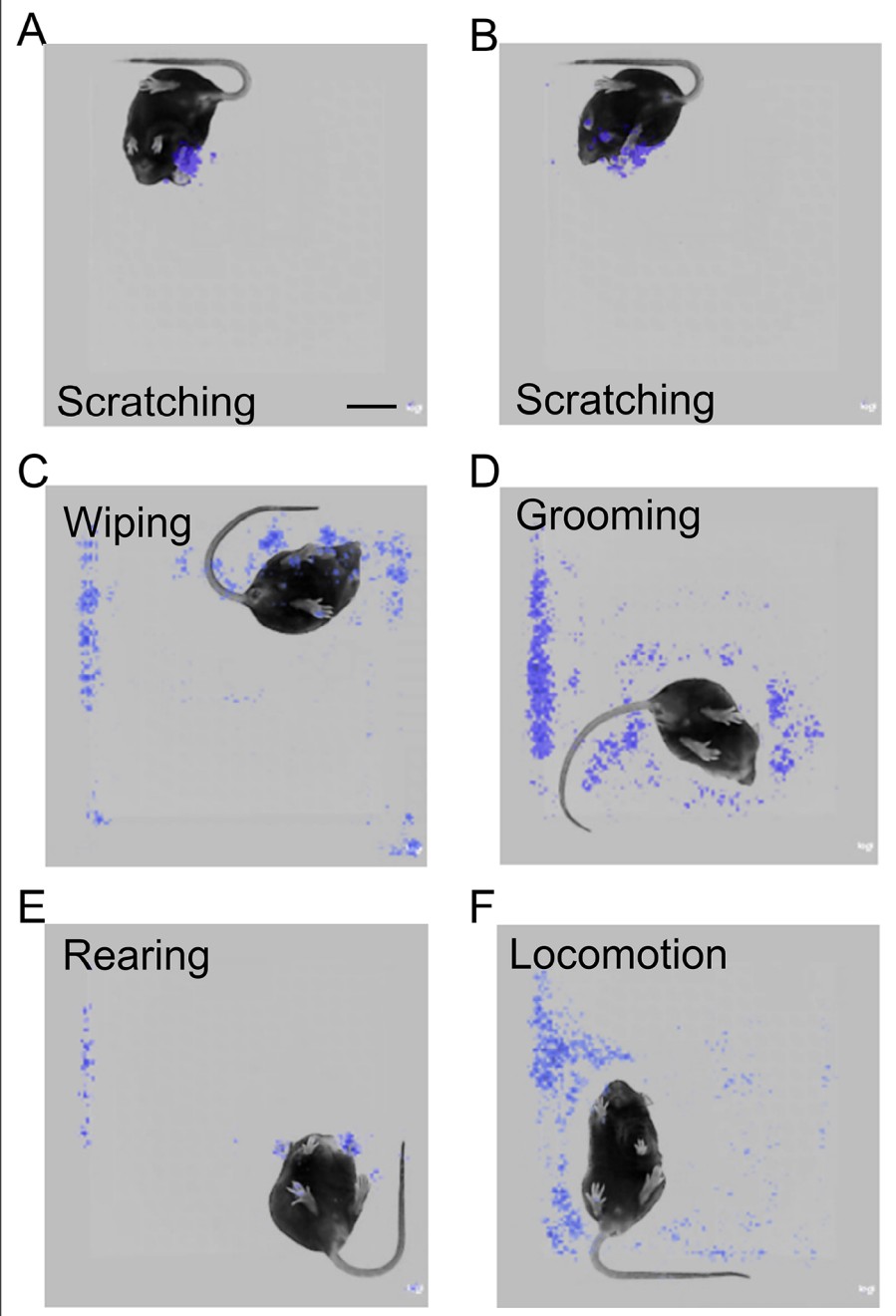

**Figure 4.** The prediction model focused on the scratching hind paw for scratching behavior recognition. (**A, B**) Saliency map showing the gradient value of each pixel of scratching frames during mouse scratching behavior prediction by the best model. The model focused on the scratching hind paw (**A, B**) and other body parts, such as front paws (**B**).Scale bar, 2 cm. Saliency map showing the gradient value of each pixel of wiping (**C**), grooming (**D**), rearing (**E**), and locomotion (**F**) frames during mouse scratching behavior prediction by the model.

The online version of this article includes the following figure supplement(s) for figure 4:

**Figure supplement 1.** Saliency map of mouse scratching and other behaviors during the prediction.

a mouse lifted its hind paw, but more ambiguous to determine when a mouse put its hind paw back onto the floor to complete a scratching train. The start and end shift did not correlate with the length of a scratching train (*Figure 3—figure supplement 5D*). Thus, the relative error (percentage of error frames) would decrease when the duration of a scratching train increases. Indeed, the prediction

accuracy (as indicated by the $F$1 score) positively correlated ($R^2$ = 0.5723) with the average scratching train duration in a video (*Figure 3—figure supplement 5E*).

Type 4 error was caused when two adjacent scratching trains were too close to each other and were predicted as one scratching train (*Figure 3—figure supplement 2F1–F3*). All missed intervals were shorter than 30 frames (<1 s) (*Figure 3—figure supplement 2F2*). Conversely, 51.4% of intervals less than 30 frames between the two adjacent scratching trains were not recognized. All intervals longer than 30 frames were recognized (*Figure 3—figure supplement 2F3*). Type 4 error was more common in manual annotation than in the model prediction (*Figure 3—figure supplement 3E1, E2*).

Type 5 error occurred when one scratching train was predicted as two or more scratching trains, separated by mispredicted intervals. The average lengths of these mispredicted intervals were around 10 frames by model prediction and around 40 frames by manual annotation (*Figure 3—figure supplement 2G1, G2* and *Figure 3—figure supplement 3F*). When reviewing these intervals, we found that more than 80% of them were within or partially overlapped with a paw licking phase (*Figure 3—figure supplement 2G3*), especially when the duration of the paw licking was more than 30 frames (*Figure 3—figure supplement 5F*). Thus, it seems likely that the model predicted some long licking frames within a scratching train as 'non-scratching'. Type 4 and 5 errors reflect the intrinsic complexity of scratching behavior. Human definitions and field consensus, such as using '2s' as the threshold between two adjacent scratching trains (*Darmani and Pandya, 2000*), would help to reduce these types of errors.

In summary, we have established a novel system combining a customized videotaping box and a well-trained CRNN neural network to automatically identify and quantify mouse scratching behavior with high accuracy. We named it as the Scratch-AID system.

## Performance of the Scratch-AID system on other major acute itch models

In addition to the nape, the other commonly used body location to induce mouse itch sensation is the cheek (*Shimada and LaMotte, 2008*). To test whether the Scratch-AID system trained by the nape CQ model could also recognize and quantify scratching behavior of the cheek model, we injected CQ (200 µg in 15 µl saline) into the cheek of five wildtype mice (three males and two females) and recorded seven videos (*Figure 5A*). We compared the scratching behavior quantification by the Scratch-AID and manual annotation. The Scratch-AID prediction had 93.4% recall, 94.8% precision, and 0.941 $F$1 score (*Figure 5B*), while those of manual quantification were 96.0%, 88.6%, and 0.919 (*Figure 5C*). The correlation between the Scratch-AID prediction and reference annotation was 0.9926, and that of the manual quantification was 0.9876 (*Figure 5D, E*). The total scratching time in individual videos from both model prediction and manual annotation were close to the reference annotation (*Figure 5F*). These results demonstrate that the Scratch-AID system can reliably identify and quantify scratching behavior triggered by acute itch sensation in the cheek.

Different pruritogens administrated at the same body location trigger scratching behavior with different dynamic features (*Wimalasena et al., 2021*). Thus, we tested whether the Scratch-AID system trained by CQ injection could recognize scratching behavior triggered by a different pruritogen, histamine. 100 µg histamine (in 15 µl saline) was intradermally injected into the nape and four videos were recorded (*Figure 5G*). The recall, precision, and $F$1 score from the Scratch-AID prediction were 96.6%, 90.91%, and 0.936 (*Figure 5H*), while those of the manual annotation were 96.3%, 80.5%, and 0.877 (*Figure 5I*). The correlation between the Scratch-AID prediction and reference annotation is 0.9707, and that of the manual quantification is 0.9895 (*Figure 5J, K*). The total scratching time in individual videos from both model prediction and manual annotation was similar to those of reference annotation (*Figure 5L*). Taken together, these data show that our Scratch-AID system, although only trained with the CQ nape acute itch model, can recognize and quantify scratching behavior of acute itch models induced at different skin locations or triggered by different pruritogens, and that the prediction accuracy of Scratch-AID is comparable to that of the manual annotation.

## Performance of the Scratch-AID system on a chronic itch model

Chronic itch is a devastating symptom and severely affects the quality of patients' life (*Yosipovitch and Fleischer, 2003*; *Yu et al., 2021*). Investigating the underlying mechanisms using the mouse model is important for developing novel therapies for treating chronic itch in various conditions. To

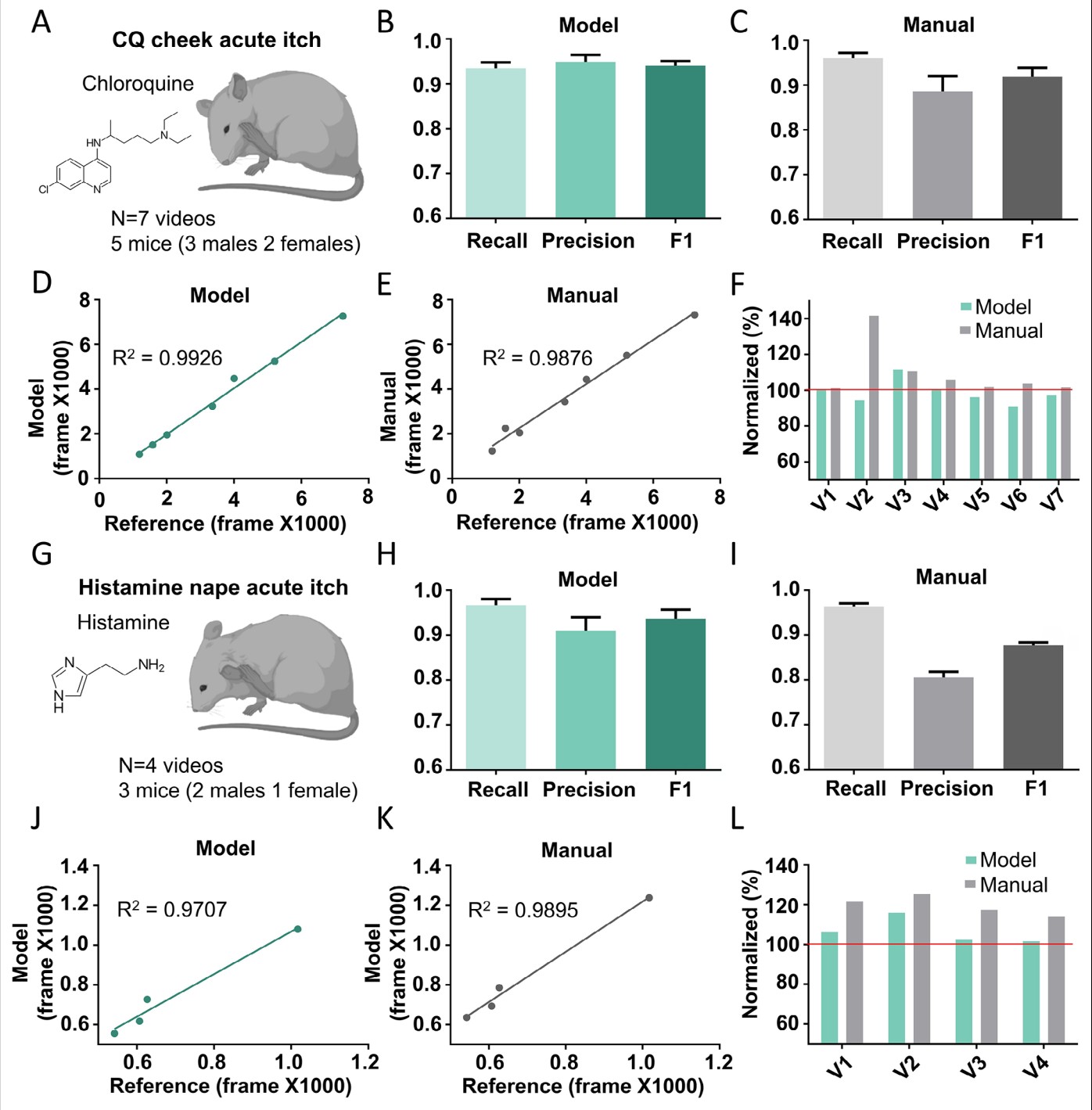

**Figure 5.** The Scratch-AID (Automatic Itch Detection) performance on other acute itch models. (**A**) A cartoon showing an acute itch model induced by chloroquine (CQ) injection in the mouse cheek. The average recall, precision, and *F*1 score of Scratch-AID (**B**) or manual annotation (**C**). Error bar, standard error of the mean (SEM). The correlation between model prediction (**D**) or manual quantification (**E**) and reference annotation. $R^2$, Pearson correlation coefficient. (**F**) The comparison among model prediction, manual quantification, and reference annotation. The reference annotation is normalized to 100% shown as the red line. (**G**) A cartoon showing an acute itch model induced by histamine injection in the mouse nape. The average recall, precision and *F*1 score of Scratch-AID (**H**) or manual annotation (**I**). Error bar, SEM. The correlation between model prediction (**J**) or manual quantification (**K**) and reference annotation. $R^2$, Pearson correlation coefficient. (**L**) The comparison among model prediction, manual quantification, and reference annotation. The reference annotation is normalized to 100% shown as the red line.

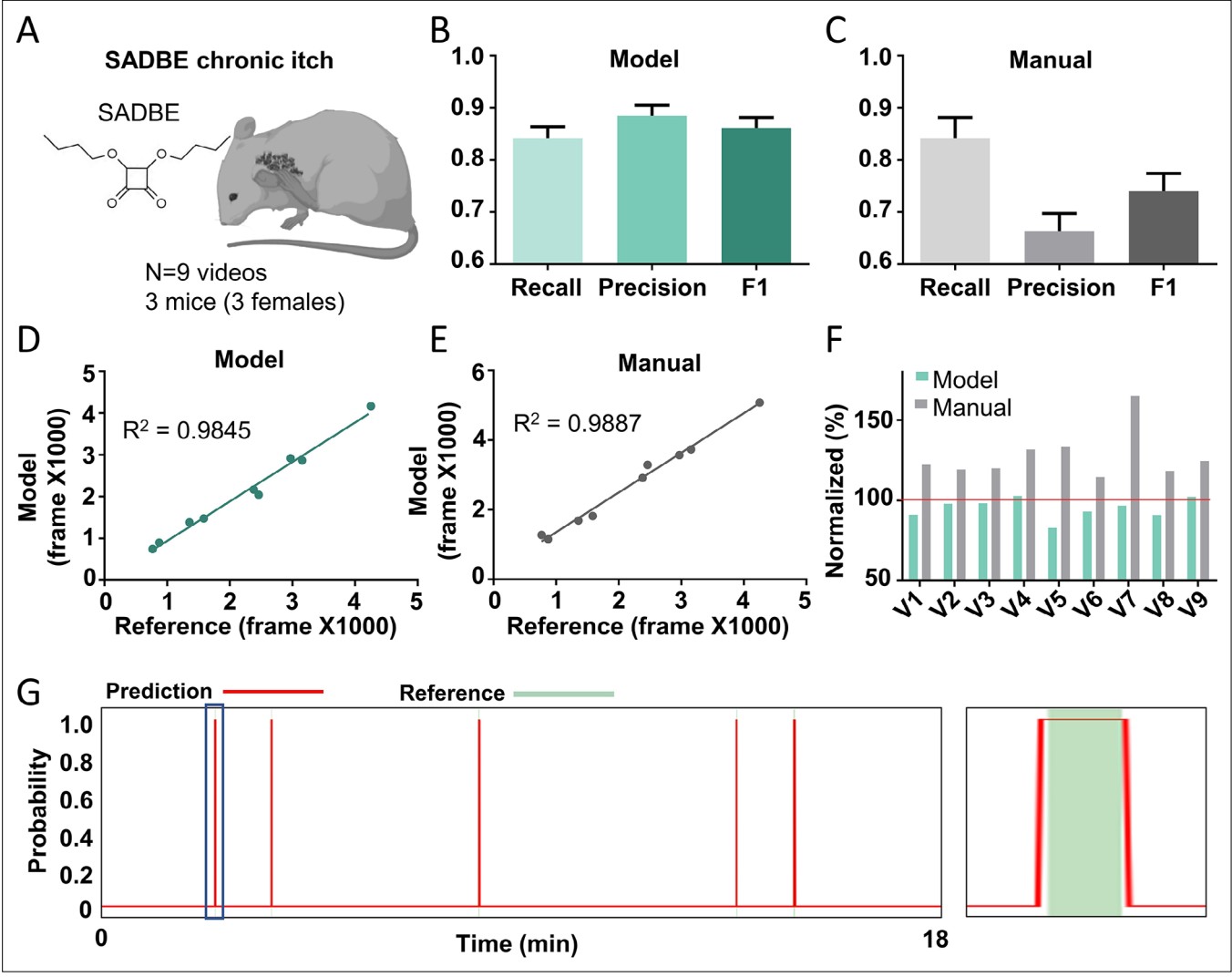

**Figure 6.** The Scratch-AID performance on a chronic itch model. (**A**) A cartoon showing a squaric acid dibutylester (SADBE) induced chronic itch model. The average recall, precision, and *F*1 score of Scratch-AID (**B**) or manual annotation (**C**). Error bar, standard error of the mean (SEM). The correlation between model prediction (**D**) or manual quantification (**E**) and reference annotation. $R^2$, Pearson correlation coefficient. (**F**) The comparison among model prediction, manual quantification, and reference annotation. The reference annotation is normalized to 100% shown as the red line. (**G**) An example scratching probability trace (red curve) predicted by the model and aligned with the reference annotation (green bar) (left). Zoom-in (right panel) of the blue square part showing nice alignment of the model prediction with the reference annotation.

The online version of this article includes the following figure supplement(s) for figure 6:

**Figure supplement 1.** Different dynamic features of chronic and acute itch models.

test whether the Scratch-AID system could be used to study mouse chronic itch models, we generated a squaric acid dibutylester (SADBE) induced contact dermatitis chronic itch model (*Beattie et al., 2022*; *Qu et al., 2015*) and recorded nine videos from three wildtype mice (*Figure 6A*). Affected mice displayed spontaneous scratching toward the nape and/or head. Noticeably, dynamic features of spontaneous scratching behavior under this chronic itch condition were different from those exhibited by the CQ acute itch model: the total scratching time was less for the same given period of time (20 min) (*Figure 6—figure supplement 1A*), and the average duration of the scratching trains was shorter (53 frames on average) than the acute scratching behavior induced by CQ (280 frames on average) (*Figure 6—figure supplement 1B* and *Figure 1F*). Despite these considerable differences, the recall, precision, and *F*1 score of the Scratch-AID prediction were 84.1%, 88.5%, and 0.862, respectively, compared to 84.1%, 66.3%, and 0.740 of manual annotation (*Figure 6B, C*). A likely reason for the decreased recall and precision of both model prediction and manual annotation was

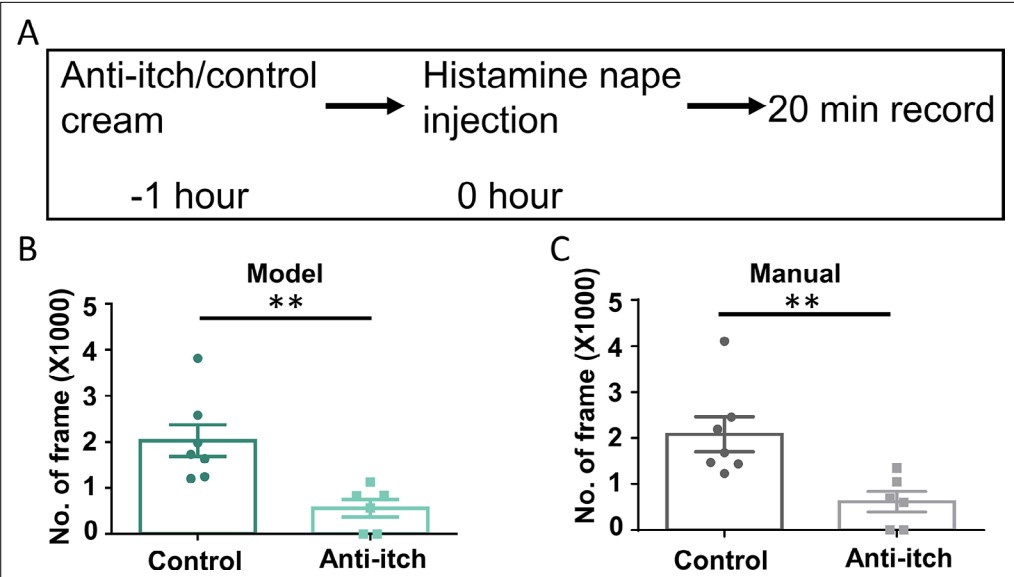

**Figure 7.** Application of the Scratch-AID (Automatic Itch Detection) system in a drug screening paradigm. (**A**) A diagram showing the experimental design of an anti-itch drug test. Quantification of scratching behavior in anti-itch cream treated group or control group by Scratch-AID (**B**) or manual annotation (**C**). Error bar, standard error of the mean (SEM). Differences between the two groups were analyzed using unpaired two-tailed Student's *t*-test, ** p < 0.01.

the dominant short scratching trains in this chronic itch model (*Figure 6—figure supplement 1B*). The correlation between the model prediction and reference annotation was 0.9845 (*Figure 6D*), which was comparable to the manual annotation 0.9887 (*Figure 6E*). The total scratching time quantified by the model was slightly more accurate than the manual annotation (*Figure 6F*). From the prediction traces (an example shown in *Figure 6G*), the Scratch-AID system was capable to capture the low-frequency short scratching trains.

### Application of the Scratch-AID system in anti-itch drug screening

Finally, we tested whether the Scratch-AID system could be applied for anti-itch drug screening. Here, we used the histaminergic acute itch nape model and the pretreatment of Benadryl (*Loew et al., 1946*), an FDA approved anti-histaminergic itch cream, as a proof-of-principle example. Benadryl or control cream was topically applied onto the mouse nape skin 1 hr before the intradermal injection of histamine (200 μg in 15 μl saline). Scratching behavior was then recorded for 20 min from six C57 wildtype mice (two males, four females) in the Benardryl treated group and seven C57 wildtype mice (two males, five females) in the control group (*Figure 7A*). Quantification of the total scratching time (frames) using the Scratch-AID showed a significant reduction with Benardryl treatment (*Figure 7B*). Similar results were found by manual annotation (*Figure 7C*). These results suggest that the Scratch-AID system is sensitive to detect the change of scratching behavior after an anti-itch drug treatment, which highlights the potential use of the Scratch-AID system in high-throughput and large-scale anti-itch drug screenings.

### Discussion

Scratching is an itch-specific behavior, and the mouse is the major model animal to study itch mechanisms and to develop novel anti-itch drugs. In this study, we developed a new system, Scratch-AID, which combined a customized videotaping box and a well-trained neural network, for automatic quantification of mouse scratching behavior with high accuracy. Its performance is comparable to the manual annotation on major itch models and an anti-itch drug screening paradigm. As far as we are aware, this is the first deep learning-based system that can achieve such high accuracy and broad generalization.

It is remarkable that a model trained with CQ induced acute nape itch videos could reliably recognize scratching behavior in other itch models, even when the body sites (nape vs. cheek) or the dynamics of scratching behavior (e.g., acute vs. chronic itch, different pruritogens) were different. The impressive performance and generalization of the Scratch-AID system are likely attributed to the high-quality and reproducible video recording; a large amount of high-quality training datasets with the frame-by-frame annotation; efficient CRNN deep learning neural network design; and training parameter optimizations.

Variable video recording condition is a major barrier that has prevented the adoption of a trained neural network by different laboratories. The trained models usually do not perform well with videos recorded under different conditions (illumination, field size, image resolution, magnification of mice, image angle, backgrounds, etc.). Thus, we built a customized videotaping box to

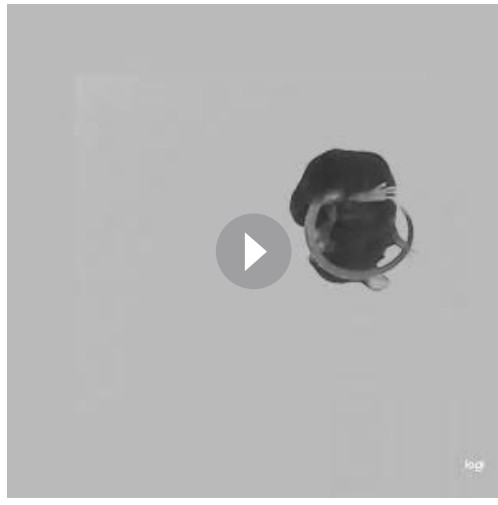

**Video 3.** An example of a rare posture during the scratching behavior with scratching hind paw partially occluded by the tail.
https://elifesciences.org/articles/84042/figures#video3

provide a reproducible and high-quality recording environment (*Figure 1B*). This helps to standardize the videotaping process, reduce the noise, and facilitate the stable performance and generalization of the trained deep learning models. Meanwhile, this video recording box is easy to be set up, scaled up, and adopted. Combined with the well-trained prediction model that has a comparable performance as manual quantification, the Scratch-AID system is ready to replace the manual quantification.

The amount and quality of training datasets are important to train a high accurate and generalized model. In general, the performance of a prediction model positively corelates with the size of the training dataset. In this study, we recorded 40 scratching videos in total and chose the CQ induced acute nape itch model, which triggered robust scratching behavior in mice, for model training and testing. The large number of scratching videos and the high amount of incidence of targeted behavior in each video provided a high-quality training dataset. In addition, the clear definition of scratching train and accurate annotation of videos, as conducted frame by frame, were also critical for training, testing, and error analysis.

CRNN is the classical deep learning neural network for analysis of animal behaviors. Here, we used ResNet (*He et al., 2016*) for the CNN part, which simplifies the neural network model by constructing a residual learning block through a shortcut connection of identity mapping (*Figure 2—figure supplement 1B*). GRU was used for the RNN part for extracting dynamic features (*Figure 2—figure supplement 1C*). Our study shows that ResNet and GRU make a good combination of deep learning architecture for analyzing animal behaviors. In addition, our network design is highly efficient that the accuracy plateau was achieved after only 10 epochs of training.

There is still some room to improve the performance of our predication models. To increase the capability of Scratch-AID to capture short scratching trains, we could train the CRNN neural network with scratching videos from chronic itch models. In addition, optimization of training parameters could also help with the improvement of prediction accuracy. For example, different error types varied when changing the input length (*Figure 3—figure supplement 2B* and *Figure 3—figure supplement 5A–C*). The Type 2 and 4 errors increased when the input length increased, while Type 1, 3, and 5 errors decreased when the input length increased. Thus, it is a trade-off to optimize and select the best input length for different itch models. Increasing the size of the training dataset may also help to get better prediction models. When checking the videos with relatively low prediction accuracy, for example, the video number 5 (V5) in the chronic itch model (*Figure 6F*), we found that the missed scratching frames could be due to a rare posture during scratching behavior, in which the scratching hind paw was partially occluded by the tail (*Video 3*). Thus, with more training videos containing some

rare scratching postures, the trained neural network models could be more 'knowledgeable' for the diversity of scratching behavior.

Quantification of animal behaviors is critical for understanding the underlying molecular, cellular, and circuit mechanisms. Compared to manual analysis, the deep learning-based automatic analysis will not only improve efficiency and accuracy, but also reduce human bias and errors. Along this direction, we developed the Scratch-AID system to achieve automatic quantification of mouse scratching behavior. Our study also provides useful insights for developing new deep learning neural network models to achieve automatic analysis of other animal behaviors.

## Materials and methods

### Mouse lines and treatments

Mice (C57BL/6J purchased from The Jackson Laboratory, RRID: IMSR_JAX:000664) were housed in the John Morgan animal facility at the University of Pennsylvania. All animal treatments were conducted in accordance with protocols approved by the Institutional Animal Care and Use Committee and the guidelines of the National Institutes of Health (Protocol No. 804886).

### Acute itch model

Acute itch models were conducted as previously described (*Cui et al., 2022*). Mouse cheeks or napes were shaved 3 days prior to experiments, and mice were placed in the videotaping box for acclimation (15 min/day for 3 days). At the day of experiment, mice were first acclimated to the videotaping box for 5 min, CQ (Sigma, C6628) (200 µg in 15 µl saline) or histamine (Sigma, H7250) (100 µg in 15 µl saline) was then intradermally injected into either the cheek or the nape, and scratching behavior was recorded for 20 min.

### Behavior recording

The mouse scratching behavior was recorded using a Microsoft laptop (Windows 10 Pro, purchased from Amazon) connecting to a web camera (Logitech C920e Business Webcam, purchased from Amazon). Logitech Capture 2.06.12 Software was used to adjust the following recording parameters: brightness 170, contrast 0, resolution 720 × 720, frame rate 30 fps. The brightness and contrast could be adjusted according to ambient light to achieve consistent illumination.

### Annotation of mouse scratching behavior

The start of a scratching train was defined as when the mouse started to lift up its hind paw and prepared to scratch at the beginning of a scratching train. For the end of a scratching train, there were two cases. If the mouse did not lick its hind paw after last scratching bout, the end frame would be when the mouse put its hind paw back onto the ground; if the mouse licked its hind paw after last scratching bout, the end frame would be when the mouse put its hind paw into the mouth. For the manual annotation, each video was manually watched at the normal (1×) speed (30 frames/s). The start and end time point of each scratching train was manually annotated, the time point was converted into the frame number (30 frames/s) for downstream analysis. For the reference annotation, each video was first converted into individual frames using python package OpenCV (*Bradski, 2000*). Then the start and end of each scratching train was determined frame by frame. Frames within a scratching train were defined as 'scratching' frames, otherwise defined as 'non-scratching' frames. For CQ cheek acute itch video number 2 (*Figure 5F* V2) and SADBE chronic itch video number 5 (*Figure 6F* V5), long lickings (>60 frames) during scratching trains were annotated as non-scratching frames for the model evaluation. For group comparison, the annotation was performed in a double-blind manner.

### Training sample preparation and preprocessing

In the training procedure, 40 videos from 10 mice were first randomly split into 32 training videos from 8 mice (V1–V32), and 8 test videos from 2 mice (V33–V40). For cross-validation, training and test videos were rotated. Test videos were V1–V8, V9–V16, V17–V24, V25–V32, or V33–V40, and the rest of videos were used as training videos. For the model training, input was prepared by the following procedure: First, each video was converted into frames; then, consecutive $N$ frames ($N$ = 3, 5, 7, 13, 23, or 45) were selected as one input with an interval of 4–10 frames between two adjacent inputs.

Then each input was annotated as 'scratching' (class 1) if more than $N/2$ frames were scratching frames, otherwise labeled as 'non-scratching' (class 0).

All frames in each input were first converted into gray scale images and resized to 300 × 300. Then a random square crop with size ranging from 288 × 288 to 300 × 300 was applied and followed by a random horizontal and vertical flip with probability of 0.5. Finally, these frames were resized to 256 × 256 and fed into the CRNN network.

## CRNN architecture

Our CRNN model (*Figure 2A* and *Figure 2—figure supplement 1*) consisted of a CNN part, an RNN part, and a full connection (FC) part. The CNN was modified from ResNet-18 (*He et al., 2016*) by changing the final FC layer into an FC layer with embedding size 256. The RNN consisted of two bidirectional GRU layers with hidden vector size 512. The FC part consisted of two FC layers with embedding size 256 and ReLu activation, and embedding size 2 for the final prediction. The prediction results were transformed into maximum value in the final output vector.

## Model training and predication

The model was trained by PyTorch (version 1.10.2, RRID:SCR_018536) (*Paszke et al., 2019*). For model training hyperparameters, batch size was set as 16 or 32 depending on the input size. The max epoch was set as 20. ADAM optimizer was used with initial learning rate $10^{-4}$ and the learning rate reduced by multiplying factor 0.3 every 5 epochs. The binary cross entropy was used as loss function. Dropout rate of FC layer was set as 0.2. The model was trained on a customized desktop with Intel i9-10900k CPU (purchased from Newegg), 64 GB RAM (CORSAIR Vengeance LPX 64 GB, purchased from Newegg), and NVIDIA GeForce RTX 3090 with 24 GB memory (purchased from Amazon).

For model prediction on new videos, the input preparation was similar to the training dataset, except that the adjacent inputs were only 1 frame apart. Each individual frame was predicted as 'scratching' or 'non-scratching' based on the following rule: the prediction of the middle frame of each input would be the same as the input prediction. For the few frames at the beginning or at the end of each video (depending on the input length $N$) that could not be the middle frame of an input, they were predicted as 'scratching' or 'non-scratching' based on the first input or last input prediction.

## Saliency map

To obtain the saliency map, we first calculated gradient value of each pixel and kept the only positive gradients, then rescaled into range 0–1 based on a previous published paper (*Selvaraju et al., 2017*). Then we generated the heatmap based on the gradient values (<0.1 transparent, 0.1–0.6 light blue to dark blue, >0.6 dark blue) and stacked the heatmap onto the original frame.

## SADBE chronic itch model

Mice (8–12 weeks) were singly housed before starting the behavior experiment. In day 1, mice were individually anesthetized in chamber using isoflurane until they did not move and showed decreased respiration rates. Mice were continued for anesthesia with a nose cone to allow access to the body of the mouse. After ensuring a mouse was fully anesthetized, the abdominal skin was shaved. In day 4, after anesthesia similar in day 1, 25 μl 1% SADBE in acetone (Sigma, 3339792) was applied to the shaved area of the abdominal skin. After application of SADBE, anesthesia continued for 3 more minutes to ensure SADBE fully absorbed before putting then back in the home cage. In days 9–12, mice were habituated in the videotaping box for 1 hr and 15 min. In day 11, the nape was shaved after habituation. In days 13–18, mice were habituated in the videotaping box for 1 hr and 15 min, then 25 μl of 1% SADBE was applied to the nape. In day 18, after 1% SADBE application, each mouse was put back into the videotaping box and recorded for three videos.

## Anti-itch drug experiment

The nape of mice (8–12 weeks, two males, five females) was shaved 3 days prior to experiments. Mice were placed in the videotaping box for acclimation (15 min per day for 3 days). At the day of experiment, mice were randomly selected and acclimated in the videotaping box for 5 min. Then, 0.2 g anti-itch cream (Benadryl, Johnson & Johnson Consumer Inc, purchased from CVS) or control cream (Neutrogena, Johnson Consumer Inc, purchased from CVS) was applied into the nape. One hour later,

histamine (200 µg in 15 µl saline) was intradermally injected into the nape, and scratching behavior was recorded for 20 min. The experimenter who did the histamine injection was blind to anti-itch cream or control cream.

### Illustration drawing

Cartoons with mice were made partially in BioRender (*BioRender, 2022*, RRID:SCR_018361).

### Statistical analysis

Data and statistical analyses were performed using Prism 6.0 (GraphPad Software, RRID:SCR_002798). The criteria for significance were: ns (not significant) $p \geq 0.05$, *$p < 0.05$, **$p < 0.01$, ***$p < 0.001$. Differences in means between two groups were analyzed using unpaired two-tailed Student's *t*-test.

## Acknowledgements

We thank the Luo and Ma lab members for their help and support. We thank Dr. Ji Zhu, Mr. Simin Liu, Dr. You Lv, Ms. Yakun Wang, and Dr. Wei Yang for their help and suggestions for this project. This work was supported by NSF grant (DMS-1854770) of Dr. Arsuaga, NIH R01 (NS083702) of Dr. Luo, and R34 (NS118411) of Drs. Ding and Luo.

## Additional information

### Funding

| Funder | Grant reference number | Author |
| --- | --- | --- |
| National Science Foundation | DMS-1854770 | Javier Arsuaga |
| National Institutes of Health | R01 NS083702 | Wenqin Luo |
| National Institutes of Health | R34 NS118411 | Long Ding Wenqin Luo |

The funders had no role in study design, data collection, and interpretation, or the decision to submit the work for publication.

### Author contributions

Huasheng Yu, Jingwei Xiong, Conceptualization, Resources, Data curation, Software, Formal analysis, Supervision, Validation, Investigation, Visualization, Methodology, Writing - original draft, Project administration, Writing - review and editing; Adam Yongxin Ye, Conceptualization, Resources, Data curation, Software, Formal analysis, Validation, Investigation, Visualization, Methodology, Writing - review and editing; Suna Li Cranfill, Conceptualization, Data curation, Software, Formal analysis, Investigation, Visualization, Methodology, Writing - review and editing; Tariq Cannonier, Conceptualization, Data curation, Visualization, Methodology, Writing - review and editing; Mayank Gautam, Conceptualization, Data curation, Methodology; Marina Zhang, Conceptualization, Software, Methodology; Rayan Bilal, Conceptualization, Data curation, Software, Methodology; Jong-Eun Park, Data curation, Software, Methodology; Yuji Xue, Data curation, Visualization, Methodology; Vidhur Polam, Data curation, Visualization; Zora Vujovic, Daniel Dai, William Ong, Jasper Ip, Amanda Hsieh, Nour Mimouni, Alejandra Lozada, Medhini Sosale, Alex Ahn, Data curation; Minghong Ma, Conceptualization, Data curation, Methodology, Writing - review and editing; Long Ding, Conceptualization, Supervision, Methodology, Writing - review and editing; Javier Arsuaga, Conceptualization, Supervision, Funding acquisition, Methodology, Writing - review and editing; Wenqin Luo, Conceptualization, Resources, Data curation, Software, Formal analysis, Supervision, Funding acquisition, Validation, Investigation, Visualization, Methodology, Project administration, Writing - review and editing

### Author ORCIDs

Huasheng Yu http://orcid.org/0000-0001-5641-2512
Suna Li Cranfill http://orcid.org/0000-0002-3431-0061

Mayank Gautam (iD) http://orcid.org/0000-0002-7257-5837
Jasper Ip (iD) http://orcid.org/0000-0001-9773-1544
Long Ding (iD) http://orcid.org/0000-0002-1716-3848
Wenqin Luo (iD) http://orcid.org/0000-0002-2486-807X

### Ethics

Mice were housed in the John Morgan animal facility at the University of Pennsylvania. All animal treatments were conducted in accordance with protocols approved by the Institutional Animal Care and Use Committee and the guidelines of the National Institutes of Health (Protocol No. 804886).

### Decision letter and Author response

Decision letter https://doi.org/10.7554/eLife.84042.sa1
Author response https://doi.org/10.7554/eLife.84042.sa2

## Additional files

### Supplementary files
- Transparent reporting form
- Supplementary file 1. Mouse information used in the recording of the training and test videos.
- Supplementary file 2. Scratching behavior summary in the 40 training and test videos.

### Data availability

The training and test videos generated during the current study can be downloaded from DRYAD (https://doi.org/10.5061/dryad.mw6m9060s). The codes for model training and test can be downloaded from GitHub (https://github.com/taimeimiaole/Scratch-AID, copy archived at swh:1:rev:d8a1e6b94e54be2c857285d74623e495a6bd47bf).

The following dataset was generated:

| Author(s) | Year | Dataset title | Dataset URL | Database and Identifier |
|-----------|------|---------------|-------------|-------------------------|
| Luo W | 2022 | Data From: Scratch-AID: a deep-learning based system for automatic detection of mouse scratching behavior with high accuracy | http://doi.org/10.5061/dryad.mw6m9060s | Dryad Digital Repository, 10.5061/dryad.mw6m9060s |

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
