## [Editor Report]

Scratch assays are the gold standard for measuring itch in rodents. However, the current limitation is that this is performed manually which is enormously taxing in terms of hours spent counting scratching bouts. The authors have developed a valuable automatic system to quantify scratch behavior with high accuracy and provided a valuable tool for the field. This will be resourceful for the greater itch biology community.

---

## [Decision Letter]

**Decision letter after peer review:**

Thank you for submitting your article " Scratch-AID: A Deep-learning Based System for Automatic Detection of Mouse Scratching Behavior with High Accuracy" for consideration by eLife. Your article has been reviewed by 2 peer reviewers, and the evaluation has been overseen by a Reviewing Editor and Catherine Dulca as the Senior Editor. The reviewers have opted to remain anonymous.

*Reviewer #1 (Recommendations for the authors):*

1. Can Scratch-AID be modified to quantify scratching bouts?

2. Saliency maps analysis suggest that the trained neural network is focusing on the movement of the hindleg to identify scratching. Although mice will spend the majority of their time scratching the injected or treated skin location, they do scratch uninjected/untreated area as well in both acute and chronic itch models. Different laboratories have different strategies to determine whether scratching to the untreated locations is included for quantification. Is Scratch-AID able to differentiate scratching directed towards treated vs untreated locations?

*Reviewer #2 (Recommendations for the authors):*

Mouse behavior assay is the primary readout for almost all itch research. The itch sensation triggers a stereotypical scratch-lick response by the ipsilateral hind paw, and several key parameters including the number of scratch bouts and total scratching duration can be precisely quantified and compared across genotypes and treatment groups. So far, the vast majority of itch behavior videos are manually scored, requiring hundreds of hours of labor and are prone to experimenter errors. In this paper, the authors introduce an automated system to replace the manual itch counting process. First, they designed a standardized videotaping box to produce consistent and high-quality videos of itch behaviors. They then annotated the videos frame by frame for scratch bouts, and trained a neural network to identify scratching behaviors with high precision. The system (named Scratch-AID by the authors) performed consistently with manual and reference scoring, and yielded key quantitative parameters (scratching bout counts and total scratching times). Saliency map revealed that the trained neural network focused on the scratching hind paw, which is consistent with what human observers look at. The authors further showed that Scratch-AID can be easily adapted to quantify scratching at slightly different sites (nape vs cheek), in response to different itch mediators (chloroquine vs histamine), in both acute and chronic itch models, and that it successfully revealed the effect of a generic anti-itch drug Benadryl. Overall, the model appears accurate and easily adaptable and can probably be widely used by the itch field.

There had been a few previous attempts to train neural networks to score itch behaviors (Kobayashi et al., 2021; Sakamoto et al., 2022). The main advance of this paper is the incorporation of a standardized recording system, and larger number of videos in the training dataset. The authors provided very detailed and thorough analyses of their own model, but did not compare its sensitivity and precision to the previous models. It is also unclear why the authors did not choose the high-speed video system that they previously published for detailed analyses of pain behaviors (Abdus-Saboor et al., 2019), despite evidence that multiple itch parameters can only be captured at high speed (Wimalasena, et al. 2021). The authors show that scratch-AID can accurately score multiple itch models (despite it being trained on chloroquine nape injection videos), but does not show whether it can distinguish itch behaviors induced by different mechanisms and mediators. For example, whether it can detect activation of different itch neuron populations (termed NP1, NP2 and NP3 by single cell transcriptomic analysis), or distinguish direct itch neuron activation by mediators such as chloroquine vs mast cell activators such as 48/80. Nonetheless, this study provides a great opportunity for the itch field to standardize behavior data acquisition and analysis, and will greatly enhance the throughput and quality of future itch behavior data.

The data are overall very convincing and the system an exciting advancement that will probably be widely used. I have some questions regarding data analysis and potential concerns for future users of the technology.

1. Does the model distinguish ipsilateral vs contralateral scratches?

2. Can the authors present the data in more detail? For example, present and compare both scratch bout count and total scratching time between scratch-AID and manual scoring for each of the itch models.

3. This is probably a given but can the authors provide evidence that the model clearly distinguishes pain (by capsaicin injection) vs itch (chloroquine and histamine)?

4. If all parameters are analyzed in more detail (average duration of scratch bouts, temporal distribution of scratches across the imaging period), can the model distinguish the differences between NP1(by β-alanine), NP2(by chloroquine) and NP3(by 5-HT) activation?

5. Can the model distinguish direct neuronal activation (chloroquine) vs immune cell activation (48/80, anti-IgE)?

6. Since the training is done with black mice, can the model detect itch behavior by white mice?

7. Can the authors comment on how the new system compares to previous itch-counting apparatus (such as SCLABA-REAL) and previous neural network models?

8. What's the max number of animals that can be recorded and analyzed in a single box?

9. Since video quality is vital for correct predictions by neural networks, can the authors comment on the influence of potential confounding factors (urine and feces, wear and tear of the acrylic imaging surface).

10. This is potentially a great opportunity for the itch field to standardize the itch behavior assay. Can the authors comment briefly in the discussion on other aspects of standardization such as time of the day for experiments (circadian rhythm), injection volumes, habituation, and sexual dimorphism?

11. Can the authors discuss how the rest of the field can apply the system in their own research? Will it be released as an open source software? And will the custom video box be commercially available?

---

## [Author Response]

Reviewer #1 (Recommendations for the authors):1. Can Scratch-AID be modified to quantify scratching bouts?

We thank the reviewer for asking the important question. Our current model cannot directly quantify scratching bouts. Nevertheless, since the high correlation between the total number of scratching bouts and total scratching time, the total scratching bouts can be estimated based on the total scratching time and average duration of each scratching bout in a specific itch model. In addition, with more detailed manual annotation of scratching bouts in each scratching train, it is possible to train a neural network to recognize and quantify scratching bouts in the future.

2. Saliency maps analysis suggest that the trained neural network is focusing on the movement of the hindleg to identify scratching. Although mice will spend the majority of their time scratching the injected or treated skin location, they do scratch uninjected/untreated area as well in both acute and chronic itch models. Different laboratories have different strategies to determine whether scratching to the untreated locations is included for quantification. Is Scratch-AID able to differentiate scratching directed towards treated vs untreated locations?

We appreciate the reviewer’s interesting comments and questions. Our current model doesn’t distinguish the scratching location. We viewed it as a strength that Scratch-AID can be generalized to recognize scratching in different body locations and itch models. This could become a problem if the number of off-target scratching makes a significant portion of the total scratching and obscures the real phenotype. In this case, the researchers may want to double check the videos and distinguish the scratching location, such as ipsilateral vs. contralateral, or cheek vs. nape. In principle, with more training videos containing different scratching locations and more detailed manual annotations, a neural network can be trained to only recognize scratching towards a particular body location.

Reviewer #2 (Recommendations for the authors):[…]The data are overall very convincing and the system an exciting advancement that will probably be widely used. I have some questions regarding data analysis and potential concerns for future users of the technology.1. Does the model distinguish ipsilateral vs contralateral scratches?

Our current models cannot distinguish ipsilateral vs contralateral scratches. In principle, with the manual annotation of left or right-side scratching, a neural network can be trained to distinguish left vs right scratching. This could be an interesting future direction.

2. Can the authors present the data in more detail? For example, present and compare both scratch bout count and total scratching time between scratch-AID and manual scoring for each of the itch models.

We systematically compared the total scratching time between scratch-AID and manual scoring for different itch models (Fig. 3, Fig. 5 and Fig. 6). Our current model cannot count the scratch bouts. As discussed above (answers to question #1 of reviewer #1), it will take additional manual annotation to train the neural network for this purpose.

3. This is probably a given but can the authors provide evidence that the model clearly distinguishes pain (by capsaicin injection) vs itch (chloroquine and histamine)?

Our model can distinguish itch-induced scratching behavior and pain-related wiping behavior. In all test videos, no wiping behavior was recognized as scratching behavior. We showed examples that the wiping behavior and other behaviors were not recognized as scratching behavior (false positive, type I error, Figure 3—figure supplement 4).

4. If all parameters are analyzed in more detail (average duration of scratch bouts, temporal distribution of scratches across the imaging period), can the model distinguish the differences between NP1(by β-alanine), NP2(by chloroquine) and NP3(by 5-HT) activation?

Our current model recognizes scratching trains. Thus, it can display the temporal distribution of scratching trains across the imaging period, quantify scratching train durations or average duration, but cannot analyze the average duration of scratching bouts (as the model can’t quantify the scratching bouts). If the temporal distribution of scratching trains or scratching train durations are different between NP1, NP2 and NP3 neuron mediated itch, our model may distinguish the differences.

5. Can the model distinguish direct neuronal activation (chloroquine) vs immune cell activation (48/80, anti-IgE)?

As discussed above, our current model can quantify the total scratching time, display the temporal distribution of scratching trains, and quantify scratching train durations. If these features show differences in scratching behavior induced by neuronal activation vs immune cell activation, for example the direct neuronal activation might trigger scratching behaviors in a shorter delay than the immune cell activation, then our model could be able to distinguish direct neuronal activation vs immune cell activation.

6. Since the training is done with black mice, can the model detect itch behavior by white mice?

We haven’t tested our model using white mice. Since the videotaping box is white color, the video quality for white mice might decrease due to the low contrast. Testing Scratch-AID with mice of different coat colors could be a future direction. If the contrast is the problem, one option is to change to box color for better contrast.

7. Can the authors comment on how the new system compares to previous itch-counting apparatus (such as SCLABA-REAL) and previous neural network models?

We appreciate the reviewer raising the question. However, it’s hard for us to do the side-by-side comparison between our model and SCLABA-REAL, because there are no published data about performance of SCLABA-REAL. According to the product information, SCLABA-REAL records the video from top view, and uses frame-to-frame subtraction to recognize the scratching behavior in a real time. Our Scratch-AID records the video from bottom view and uses raw frames to recognize the scratching behavior in an off-line manner.

Compared to previous neural network models, the main novelties of our study include: (1) designed of a videotaping box for stable and reproducible video recording in different labs; (2) collected a large amount of frame-labeled dataset for the training; (3) optimized CRNN architecture and hyperparameters for more effective learning; and (4) tested the generalization of the model in major mouse itch models and in a drug screening paradigm.

8. What's the max number of animals that can be recorded and analyzed in a single box?

Only one mouse can be recorded and analyzed in a single box. However, with multiple boxes, several mice can be recorded simultaneously.

9. Since video quality is vital for correct predictions by neural networks, can the authors comment on the influence of potential confounding factors (urine and feces, wear and tear of the acrylic imaging surface).

This is a very good question. According to our experience, sufficient habituation greatly reduced the urine and feces. We also cleaned the floor (urine and feces) between two recordings. A small amount of urine and feces did not have significant influence on the performance of the model. We don’t think that wear and tear of the acrylic imaging surface will be a big issue. After around 200 video recordings, no obvious wear and tear of the acrylic imaging surface were observed. Even if there are some wear and tear after long-term use, it’s easy to replace the imaging floor with a new one (same material and quality).

10. This is potentially a great opportunity for the itch field to standardize the itch behavior assay. Can the authors comment briefly in the discussion on other aspects of standardization such as time of the day for experiments (circadian rhythm), injection volumes, habituation, and sexual dimorphism?

The reviewer raises an important point. As the reviewer mentioned, circadian rhythm, injection volumes, habituation, sexual dimorphism and other factors might affect the itch behavior assay and results. For our experiments, we did the scratching behavior recording during the daytime, injected 15 ul pruritogens dissolved in saline for both cheek and nape model, habituated each mouse in the recoding box 15 min/day for three consecutive days, and included both male and female mice. To standardize the mouse itch behavior assay, the itch field need to discuss together to reach a consensus.

11. Can the authors discuss how the rest of the field can apply the system in their own research? Will it be released as an open source software? And will the custom video box be commercially available?

We really hope that the rest of the mouse itch field will try and take advantage of our Scratch-AID system for their studies. This is the motivation for us to work on this project. Our videos, codes, and models will be publicly available as this manuscript is published. We would also like to further optimize our prototype models, add features as suggested by the reviewers, and launch a user-friendly software or a web-based analysis system (if we could have funding support and identified a software developer).